# Comparing Condensed and Hydrolysable Tannins for Mechanical Foaming of Furanic Foams: Synthesis and Characterization

**DOI:** 10.3390/molecules28062799

**Published:** 2023-03-20

**Authors:** Jonas Eckardt, Thomas Sepperer, Emanuele Cesprini, Primož Šket, Gianluca Tondi

**Affiliations:** 1TESAF Department, University of Padua, Viale dell’Università 16, 35020 Legnaro, Italy; 2Department of Green Engineering and Circular Design, Salzburg University of Applied Sciences, Marktstraße 136a, 5431 Kuchl, Austria; 3Slovenian NMR Center, National Institute of Chemistry, 1000 Ljubljana, Slovenia

**Keywords:** bio-polymers, bio-based materials, lightweight, natural extracts, insulation, ^13^CNMR analysis, ATR FT-IR

## Abstract

This study examined the potential of hydrolysable tannin in comparison to condensed tannins for the production of furanic foams. The results indicate that chestnut tannin presents lower reactivity and requires a stronger acid for the polymerization. Additionally, foamability and density were found to be dependent on both surfactant concentration and tannin type, allowing lower densities for mimosa tannin and lower thermal conductivities for chestnut-based foams. Mimosa tannin was found to have the highest compression strength, followed by quebracho and chestnut, promising thermal conductivity of around 50 mW/m·K for 300 kg/m^3^ foams, which suggests that chestnut foams have the potential to performing highly when the density is reduced. Chemical analysis revealed that the methylene moieties of the furanics are non-specific and produces new covalent bonds with nucleophilic substrates: -OH groups and free-positions in the flavonoids. Overall, this study opens new perspectives for the application of hydrolysable tannins in polymer and material science.

## 1. Introduction

At present, the market for insulation materials is dominated by oil derivate products such as polyurethane, extruded polystyrene (XPS) and expanded polystyrene (EPS) or mineral wool [1,2]. Natural insulation materials tend to perform more favorable in terms of their environmental impact [3]. However, products such as mineral wool are reported to possibly have a stronger negative environmental impact than EPS [4]. The market for bio-based insulation materials is increasing, but their market share still corresponds to a marginal fraction of the global insulation market, mostly due to their higher economic cost [5]. Being aware of this issue, the scientific community is increasingly looking for high-performing green alternatives in this field, suggesting the use of wood extractives rich in tannin to produce bio-based foams that can potentially be used as insulation material.

Tannin foams are highly porous materials with low thermal conductivity and good fire resistance [6] that can be obtained by acid/alkali catalysed polymerization of tannin with different crosslinkers such as formaldehyde, hexamine, furfuryl alcohol or by creating urethanes with and without isocyanates [7,8,9,10]. Furthermore, they could be used as acoustic absorbers, show high resistance to acids bases and solvents [8], and could potentially be used for wastewater treatment due to their capability to filter pollutants such as dyes, anionic detergents and several pharmaceutical compounds [11,12]. After the first formulations [13], many enhancements were completed in terms of greener and safer formulations [14,15,16], and towards more performing insulative and mechanical properties [9,17,18,19,20]. Additionally, several production processes were developed, which showed the high relevance of the process and the formulation for the resulting material properties [21,22,23]. In this context, many different formulations were developed using not just different crosslinkers, additives and production methods, but also different types of tannin. Tannins consist of a complex polyphenolic structure that can be found in many plants. They are namely known to be used in leather tanning, wood and other adhesives, fisheries, beverage manufacturing and animal feed. It can be obtained via water extraction from different parts of plants. Depending on the material used for tannin extraction, either the bark, wood, leaves or seeds can be used [24,25].

Tannins can be classified into hydrolysable and condensed. Condensed tannins are oligomers of repeating monoflavonoid units, with amino and imino acid traces. For a tannin to be classified as condensed, three to eight flavonoid repetition units are needed [26]. The main flavonoids in mimosa tannin are fisetinidin and robinetinidin (about 87%), followed by catechin and gallocatechin [27]. In addition, quebracho tannin extract is specifically composed of fisetinidin and robinetinidin, with a slightly higher proportion of catechin/epicatechin units, and was found to be slightly more polymerized than a mimosa tannin extract [28]. These flavonoids are particularly reactive especially in the o-positions of the A ring. Hydrolysable tannins are esters of simple sugars with gallic and ellagic acid. The main constituents of chestnut tannin are castalagin and vescalagin followed by vescalin and castalin, gallic acid and pentagalloyl glucose [29]. The hydrolyzable tannin’s aromatic rings (namely gallic or ellagic acid) were found to be much less reactive (as monomer and bound in natural oligomers) compared to phenol, as only the much less reactive meta-sites remain free, but also in the comparison with condensed tannins [30].

Tannin foams can be easily produced using condensed tannins (mimosa, quebracho, pine and spruce) [12,31,32]. Conversely, hydrolysable tannins are much less reactive and little research was conducted on their use for polymerization processes. Though a partially usage in a phenolic system, using tannic acid and furfuryl alcohol as the main building blocks and a urethane-based approach already revealed that hydrolysable tannins also have some potential for foam production [30,33,34,35]. Since several parameters differ in the various tannin foam formulations found in research, it is difficult to compare the influence of different types of tannins on foam properties. This research therefore aims to compare different commercially available tannin extracts in the production of mechanically blown furanic foam formulation. Specifically, the comparison will involve two condensed tannins, namely mimosa and quebracho (*Acacia mearnsii*, *Schinopsis* sp.), as well as a hydrolysable tannin extracted from chestnut (*Castanea sativa*).

Considering the complexity of the system, in this article we also observed the effect of different surfactant concentrations to give a broader overview on the topic.

## 2. Results and Discussion

The foams, subject of the present study, were characterized for their physical and chemical properties. In Figure 1, the different appearances of the three foams with different tannin extracts produced at a surfactant concentration of 10% can be seen.

Producing foams with the exact same formulation by just changing the tannin type was not possible due to the different reactivity of condensed and hydrolysable tannin. For instance, using 2 mol/L sulphuric acid in combination for condensed tannins is effective, whereas for chestnut the foam collapsed before curing.

On the other hand, using 3 mol/L sulphuric acid is effective with chestnut tannin, but the curing occurs too fast (already in the mixing pot) for mimosa or quebracho. This hinders the preparation of highly porous foams. It is known that higher curing temperatures and increasing the catalyst amount can accelerate the curing time of tannin foams [36,37]. Preliminary trials with chestnut tannin also showed that too high curing temperatures are problematic in terms of drying cracks and the increased activity of water as a blowing agent. Using a concentration of 4 mol/L without increasing the temperature further resulted in curing of the resin already in the mixing pot. At 3 mol/L and 90 °C, foam collapse could be avoided. In order to obtain comparable foam material and address differences in reactivity, the mimosa and quebracho foams in this study required using a concentration of 2 mol/L, whereas the production of the characterized chestnut foams required a concentration of 3 mol/L. Furthermore, at 1% surfactant concentration, only the combination with mimosa resulted in a highly porous material and was therefore the only one used for further evaluation. The main physical characteristics of the produced foams are reported according to the tannin type and surfactant amount in Table 1.

As expected, the foams produced with different tannin precursor present a broad range of features. Mimosa’s foams are lighter, quebracho’s are heavier and stronger, and chestnut’s are more homogeneous and more anisotropic. Changing the amount of surfactant showed similar trend regardless of the tannin involved, and a higher amount of surfactant led to lighter foams. Research on mechanical foaming clearly highlights the complexity and multiple factors involved in the formation of cell shape/size and foamability. These factors involve namely surfactant concentration and its relation to the critical micellar concentration, the dynamic surface tension and the surface mobility, as well as the mixing conditions [38,39]. One of the most important functions of the surfactant in foaming a resin is the formation of a stable emulsion by reducing the surface tension at the interface between the resin and the incubated air [40]. It is known that the mixing speed and time both influence the cell size and density [41]. Comparing the results to similar formulations for mimosa tannin, the foams presented show a higher pore size at similar densities (170 kg/m^3^ −81 µm compared to 162 µm), which suggests bigger cell walls for the produced foams. Aside of different rotation speed and time, the herein investigated foams also contained diethylene glycol, which seems co-responsible for the increased cell size [42]. Diethylene glycol was used as plasticizer and allows to smoothen the drying cracks during the curing. Several studies have shown that the amount of surfactant can hardly be correlated (linearly) with cell size, density or foamability, and that the optimum has to be found for each formulation. [22,40,43,44] Lower orthotropicity was observed when compared to lightweight tannin furanic foams produced with a blowing agent, which is in line with the results of a previous study that also employed a mechanical foaming system [21,42]. In this context, it should also be noted that the use of a larger container while retaining the same stirrer head led to higher densities in the production of the 30 × 30 × 5 cm^3^ foams with similar formulations.

The results of the thermal conductivity measurements for chestnut and quebracho at 10% surfactant, as well as for mimosa tannin (surfactant 1, 5, 10%), are shown in Figure 2.

The average thermal conductivity ranges between 46 and 55 mW/m·K which is comparable to foams with similar densities with and without a blowing agent (140–200 kg/m^3^–44–55 mW/m·K; 171 kg/m^3^–45 mW/m·K) [42,45]. Tannin foams with lower density that are produced with blowing agents range from around 30 mW/m·K [10,21,46]. Considering the high densities, the chestnut foams appear to have slightly lower thermal conductivity which could be due to the high homogeneity and with the almost anisotropic shape of the cells.

The compression resistance of the different tannin foams is visualized in Figure 3.

Several studies already showed the strong correlation between density and compression resistance, but also highlight the importance of the initial formulation and cell shape [10,21,46]. This correlation can clearly be found within all the groups (sig. < 0.001) (R^2^: mimosa 0.988; quebracho 0.989 chestnut 0.872). Furthermore, the influence of the tannin type is also clearly visible, having the highest compression strength for mimosa, followed by quebracho and chestnut. Comparing the results of the mimosa foams to the literature where solely FOH is used as crosslinker shows much higher compression resistance for all the herein produced foams [45]. A further comparison with tannin foams of similar densities found in the literature is summarized in Table 2.

The mimosa foams of this work even have comparable compression strength of pine, mimosa or quebracho tannin furanic foams using Formaldehyde (CH_2_O) as additional crosslinker, which is known to improve compressive strength [10,46,47]. Isocyanate-free polyurethane foams (NIPU) made from mimosa and chestnut have a much lower compressive strength at comparable density than the various tannin foams of the presented work [7,33]. Aside of the difference in usage of tannin, comparing these foams to others in the literature suggests that the foaming method in combination with the formulation results in improved mechanical stability without using an additional aldehyde. Compared to a similar mimosa tannin formulation with a mechanical foaming approach, also shows that the introduction of diethylene glycol together with a lower mixing speed and shorter time, as well as improved mechanical strength (171 and 175 kg/m^3^ -> 0.843 and 1.33 MPa) [42]. Other studies comparing different types of tannin found rather less strong differences between the mechanical performance of mimosa and quebracho tannin [48]. The deformation under stress at similar density also shows this difference, visible in Figure 4 for three different tannin type foams.

Not just a higher compressive strength but also a different elasticity for the different types of tannins can be seen. The highest compressive strength can be observed for mimosa, while also showing the highest modulus of elasticity, followed by quebracho and chestnut.

The results for the leaching test, together with the calculated recovered catalyst amount, are presented in Figure 5:

Between 12.6 and 19.8% of weight loss for the different foam formulations was observed after leaching, which was the lowest for quebracho and the highest for chestnut. If the tannin becomes fully incorporated into the polymer, slightly more than 20% would hypothetically be leachable, considering that diethylene glycol, the acid catalyst and the surfactant are not becoming a part of the polymer. Comparable works with mimosa showed slightly lower leaching (6–12%) of the polymer [42]. Quebracho formaldehyde polymers from the literature responded to a leaching of 20% [49]. The calculated acid recovery was the lowest for mimosa and the highest for chestnut, ranging between 13.0 and 26.2%. This results is in agreement with the previous studies, reporting a catalyst recovery between 7 and 39% [21,42,50].

The three different foams were analyzed through solid state ^13^C-NMR after leaching and curing and the spectra are presented in Figure 6.

The spectra are presented to highlight if and how the tannin and the furanic precursor produce new bonding or modify their chemical structure after curing.

The most interesting evidence that can be highlighted in all spectra of foams, is the presence of a new signal at 62 ppm. According to the chemical shift prediction, the signal at 62 could be due to -CH_2_- groups between the furanic ring and the oxygen of phenolic rings. Some decrease in the band at around 116 and 104 ppm can also be noticed, which is due to the free positions of the A and B aromatic rings. This would suggest a crosslinking on the free position of the aromatic ring that cannot be confirmed because the bonding signal should be at around 33 ppm and this area is occupied also by the precursors (CH_2_ bridges of PFA and the CH_2_ in the C ring of flavonoid). The reaction of condensed type of tannin with furfuryl alcohol was already subject to several studies, finding the crosslinking reaction appearing through methylene bridges -CH_2_- at C6 and C8 position, becoming visible at 116 ppm, which is also overlapping with the signal of the interflavonoid linkage C4-C6 according to the literature. This peak can be calculated at 110 ppm which is also reported as the interflavonoid linkage of C4–C8, showing the difficulty of distinguishing this area through NMR analysis alone [26,27,51].

In the quebracho foam spectrum, there is a significant increase in the signal at around 72 ppm, which corresponds to the region of carbohydrates. Additionally, there is a small decrease in the band at around 104 ppm, as previously observed in the mimosa foam profile. Furthermore, there is information available that enables us to consider bonding through free aromatic positions in this case as well.

In the chestnut foam spectrum, it can be observed that the furanic contribution is higher, but here the bonding through the free aromatic ring is probably not occurring because of the higher steric hindrance. An example of the possible crosslinking reactions that occur between furfuryl alcohol, the condensed tannin flavonoid units and gallic acid, which is the main phenolic compound in chestnut tannin, is shown in Figure 7.

Although ATR FT-IR analysis of the tannin-furanic polymers was also performed, the proposed methylene bridges between furanic and polyphenols could not be identified due to overlapping signals with the precursors. These spectra confirm the presence of carbonyl groups derived by the ring opening of the furanic (and the esters of the hydrolysable tannins) when they polymerize visible in the peak around 1710 cm^−1^ and the typical bands of phenolic moieties become visible at 1600–1610 and 1500–1520 cm^−1^ [14]. The spectra of the different foams are visualized in Figure 8.

## 3. Materials and Methods

### 3.1. Materials

Industrial mimosa tannin extract (Weibul AQ) was provided by the company Tanac S.A. (Montenegro, Brazil). Chestnut (Fintan C) and quebracho (Fintan QSF) tannin extracts were supplied by Silvateam S.p.A. (S. Michele Mondovì, Italy). According to the manufacturer, mimosa is extracted using bark, whereas quebracho and chestnut tannin is extracted from wood. Furfuryl alcohol—FOH (98%, technical) was purchased from Transfurans Chemicals (Geel, Belgium) and other chemicals (Diethylene glycol—DEG, sulfuric acid, Tween 80) were purchased from VWR (Darmstadt, Germany). Sulfuric acid was diluted to a 2 mol/L solution for the quebracho and mimosa formulation and to 3 mol/L for the chestnut formulation.

### 3.2. Method

#### 3.2.1. Foam Synthesis

The main procedure of preparing the foams was described previously in another work [42]. In general, foams were produced by mechanical agitation using an IKA (IKA, Staufen, Germany) overhead stirrer with a butterfly stirring head. In a first step, furfuryl alcohol, diethylene glycol, water and Tween 80 were mixed together. Secondly, the tannin was added and homogenized by hand for 30 s and for another 5 min using the mechanical stirrer at 200 rpm. In a last step the acid was added, and the stirring speed was increased to 1100 rpm, mixing for further 15 min. After obtaining a stable wet-foam-mass, the mixture was transferred into a PTFE-coated mold and cured in a convection oven at 70 °C for 30 min (Quebracho, Mimosa) and at 90 °C for 90 min (Chestnut) before it was removed from the mold to peel of the skin and to allow for even gaseous evaporation. Final drying was performed for another 24 h at 70 °C. The surfactant (S) concentration was chosen at 1, 5 and 10% in relation to the tannin–furfuryl alcohol (T-F) amount (S/T-F). The weight percentage of the different constituents used in the formulations is shown in Table 3. For the measurement of the compression resistance and the cell dimension, foams with the dimension 15 × 15 × 5 cm^3^ were produced, whereas foams with the dimension 30 × 30 × 5 cm^3^ were produced for the thermal conductivity measurements.

#### 3.2.2. Bulk Density

To calculate the bulk density (d) the outer and less homogeneous area (0.5 cm) of each foam was removed and cut into rectangular shape. By measuring the length (l), width (w), height (h), and weight (m) for each obtained sample, the density (ρ) was calculated according to the equation ρ = m/(l × w × h).

#### 3.2.3. Cell Dimensions and Orthotropicity

Measurements of cell size dimensions were conducted using a reflected light microscope (Nikon SMZ 1500, Tokio, Japan). At least 50 measurements in length and 100 in width were performed in the direction parallel to the growth on three samples per formulation. The average cell diameter (Cell Ø) was calculated using the following equation, where D represents the average of 150 measurements: Cell Ø = (π/4) × D. To calculate orthotropicity, the length measurement for each cell was divided by its width measurement.

#### 3.2.4. Thermal Conductivity

Thermal conductivity was assessed on samples measuring 25 × 25 × 3 cm^3^ using a λ-Meter EP500e instrument (Lambda-Messtechnik GmbH, Dresden, Germany). Measurements were conducted at three different temperatures (10, 25, 40 °C), with a 10 K temperature difference applied to the upper heating plate.

#### 3.2.5. Compression Strength

The universal testing machine Zwick Roell Z250 (Zwick-Roell, Ulm, Germany) was used to measure the compression strength on samples 5 × 5 × 4 cm^3^, following the standard ISO 844:2007 [52]. For each formulation, five specimens were tested parallel to the growth direction at a rate of 2 mm/min. If a clear break could be observed within 10% of specimen deformation, that value represented the maximum compression strength. If no clear break occurred within 10% of the specimen’s deformation the value at 10% was taken as the result.

#### 3.2.6. Acid Recovery and Leaching

Following pulverization and drying at 100 °C, 7.5 g of dry tannin foam powder was added to 100 mL of water and continuously stirred for 6 h. Subsequently the solution was filtered using a paper filter with a pore size of 25 µm. To determine the amount of recovered acid, the pH-value of the solution was measured with a WTW INOLAB pH 720 m instrument (Xylem, New York, NY, USA). The amount of recovered sulfuric acid was indirectly calculated through the pH, considering the maximum possible H^+^ concentration in 7.5 g of the polymer. The process was performed in triplicate. Furthermore, the filtered powder was dried and weighted afterwards to determine the amount of leached material by its weight %.

#### 3.2.7. Attenuated Total Reflectance Fourier-Transform Infrared (ATR FT-IR) Spectroscopy

The pulverized foams were washed with water before being dried and then scanned using a Perkin Elmer Frontier FT-IR spectrometer (Perkin-Elmer, Waltham, MA, USA), equipped with an ATR Miracle accessory. For each sample, thirty-two scans were performed at a resolution of 4 cm^−1^ and in the range of 4000 to 600 cm^−1^. For each formulation, five different samples were scanned. The obtained spectra were normalized and averaged using the SpectraGryph 1.2 software.

#### 3.2.8. ^13^C-NMR Solid State

^13^C-CP-MAS NMR spectra of solid samples were recorded on a Bruker Avance NEO 400 MHz NMR (Bruker BioSpin, Rheinstetten, Germany) spectrometer equipped with 4 mm CP-MAS probe. All samples were spun at the magic angle with 10 kHz at 25 °C for the different measurements. All spectra were analyzed using Bruker TopSpin 4.2 software.

## 4. Conclusions

This study investigated the use of different types of condensed and hydrolysable tannins (mimosa, quebracho, chestnut) for producing tannin furanic foams, by varying the surfactant amount with the method of mechanical foaming. The results showed that chestnut tannin has lower reactivity compared to condensed tannins and requires a stronger acid to reach hardening before the wet foam structure collapses. Foamability and density were found to be related to surfactant concentration and behaved different for each type of tannin, reaching the lowest densities for mimosa. Considering the density, thermal conductivity was slightly lower for chestnut tannin and quite high for tannin foams in general due to their density, but all values were in line with the literature at similar densities. The compression resistance was the highest for mimosa, followed by quebracho and the lowest for chestnut. Chemically, it was observed that the methylene moieties of the furanic are non-specific and produce new covalent bonds with nucleophilic substrate: -OH groups and free-positions in the polyphenols. However, the free aromatic positions of hydrolysable tannins are sterically hindered, and this contributes to their lower reactivity. These results suggest that the production system and formulation ingredients are determinant in producing foams with good properties, particularly when hydrolysable tannins are used, which require a specific production process to cope with its lower reactivity.

## Figures and Tables

**Figure 1 molecules-28-02799-f001:**
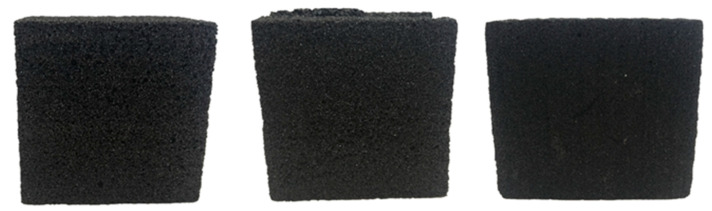
Left to right: Mimosa, quebracho, chestnut foam produced with 10% surfactant.

**Figure 2 molecules-28-02799-f002:**
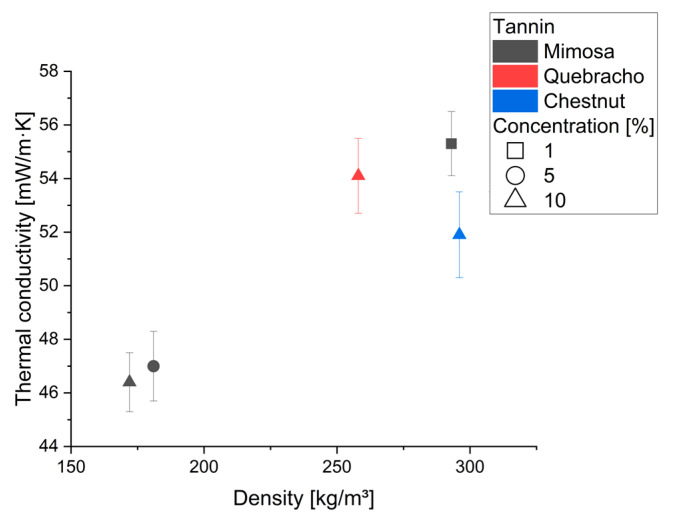
Thermal conductivity for the foams produced with mimosa at 1, 5, 10% surfactant concentration and quebracho/chestnut at 10%.

**Figure 3 molecules-28-02799-f003:**
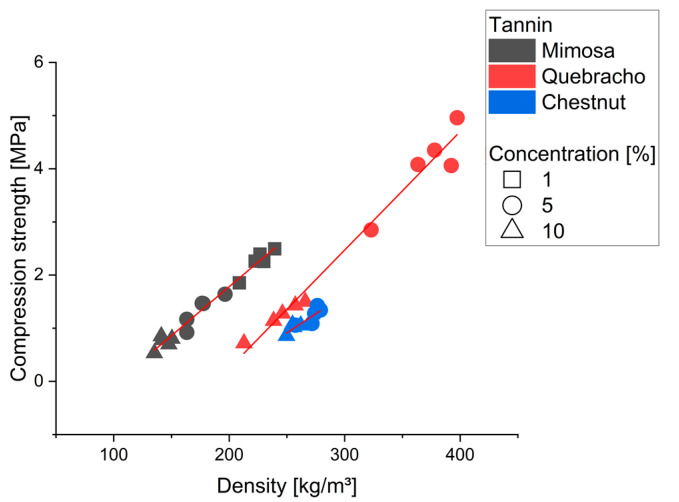
Compression strength of the different formulations according to their density.

**Figure 4 molecules-28-02799-f004:**
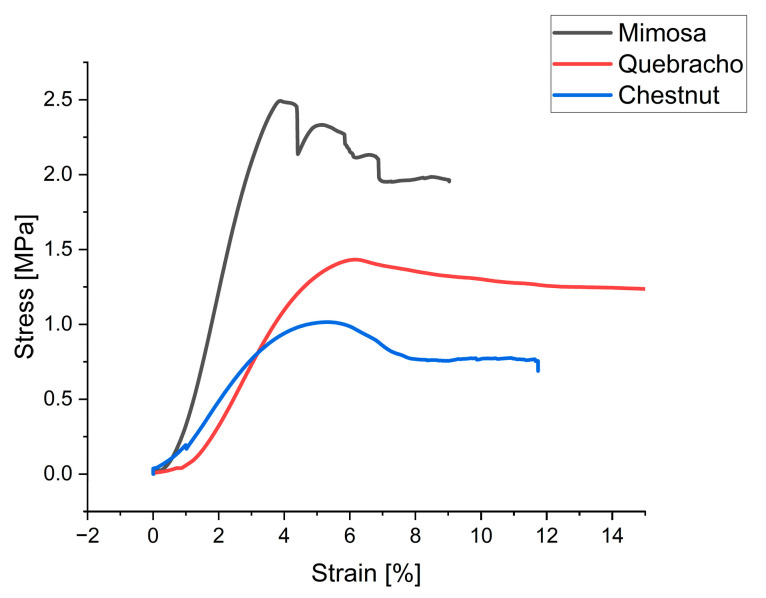
Mimosa—1% (239 kg/m^3^); quebracho—10% (257 kg/m^3^); chestnut—10% (255 kg/m^3^).

**Figure 5 molecules-28-02799-f005:**
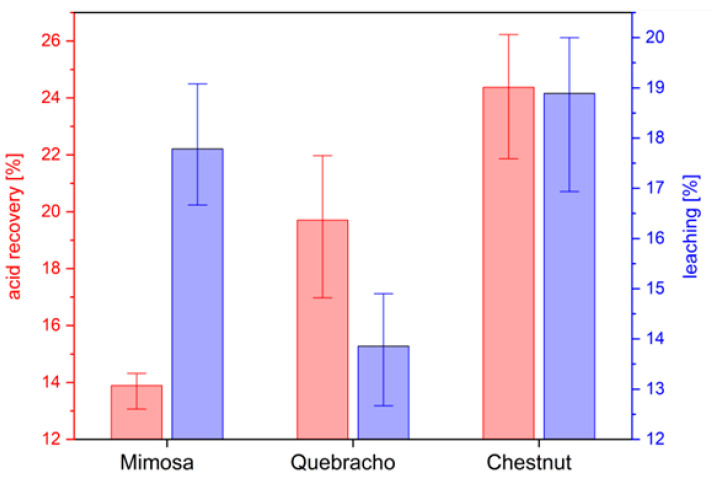
Percentage of recovered acid and leached polymer for the different tannin foam formulations produced at 5% surfactant concentration.

**Figure 6 molecules-28-02799-f006:**
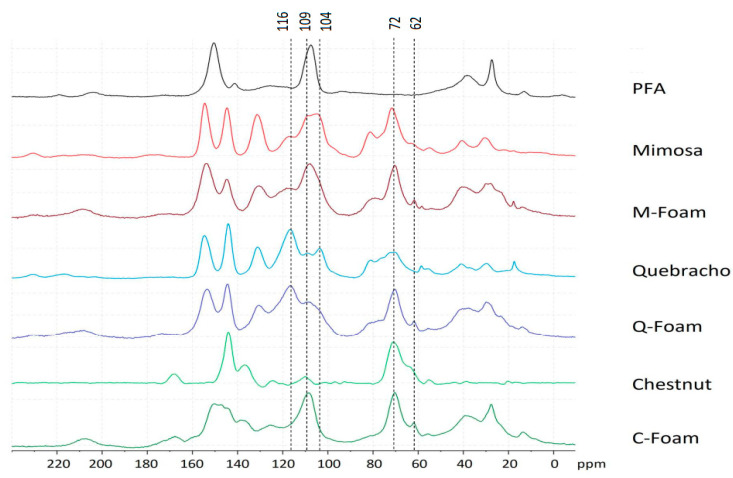
^13^C-NMR spectra of polyfurfuryl alcohol, mimosa, quebracho and chestnut tannin extract and their derivative foams.

**Figure 7 molecules-28-02799-f007:**
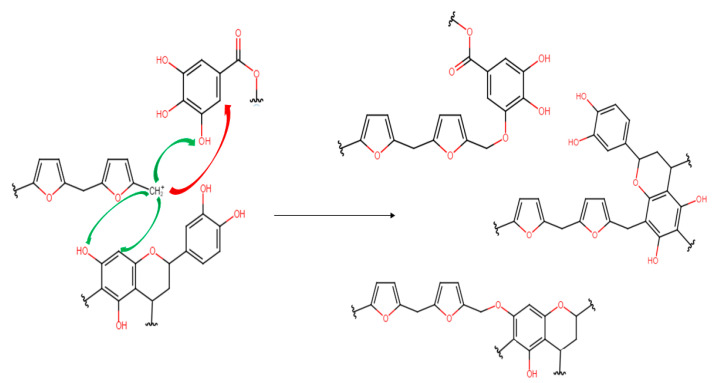
Possible mechanism of non-specific polymerization between polyfurfuryl alcohol and tannins.

**Figure 8 molecules-28-02799-f008:**
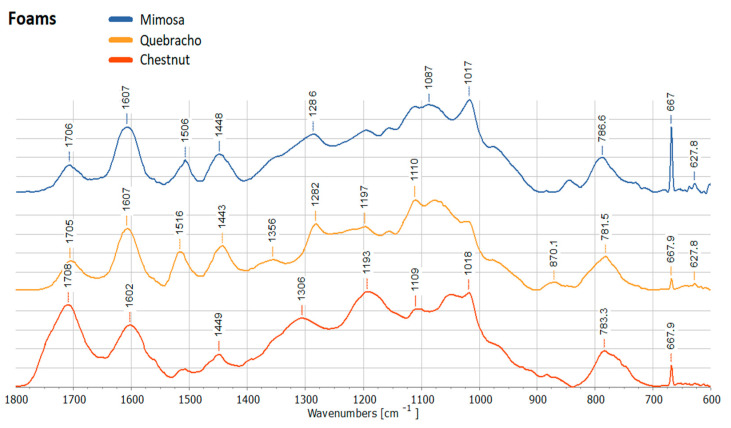
ATR FT-IR spectra of the mimosa, quebracho and chestnut-based tannin foams.

**Table 1 molecules-28-02799-t001:** Summary of main physical properties-standard deviation in brackets.

Tannin Type	Surfactant [%]	Density [kg/m^3^]	Cell Ø [µm]	Orthotropicity	Comp. Strength σ_c_ [MPa]
Mimosa	1	225.5 (10)	129 (64)	1.17 (0.09)	2.250 (0.218)
	5	175.4 (12.2)	162 (99)	1.19 (0.12)	1.334 (0.255)
	10	143.3 (5.3)	178 (113)	1.22 (0.13)	0.734 (0.112)
Quebracho	5	370.8 (26.7)	60 (28)	1.17 (0.09)	4.060 (0.687)
	10	244.1 (18.2)	117 (72)	1.14 (0.08)	1.214 (0.276)
Chestnut	5	271.6 (7.6)	103 (44)	1.11 (0.07)	1.239 (0.143)
	10	258.4 (6.8)	105 (58)	1.14 (0.09)	1.019 (0.083)

calculated cell diameter **Ø**; compression strength **σ_c._**

**Table 2 molecules-28-02799-t002:** Literature comparison of tannin-based foams with densities between 90 to 350 kg/m^3^.

Foam Formulation	Compression Resistance [MPa]	Density [kg/m^3^]	Reference
Mimosa-FOH-CH_2_O	1.04	136	[46]
	1.38	161	
	3.97	306	
Pine-FOH-CH_2_O	1.03	140	[10]
	1.75	190	
Mimosa-FOH	0.15	140	[45]
	0.5	200	
Mimosa-NIPU	0.15	150	[7]
	0.57	260	
Chestnut-NIPU	0.91	350	[33]
Tannin acid-FOH	0.1	150	[35]
Quebracho-FOH-CH_2_O	0.24	90	[47]

**Table 3 molecules-28-02799-t003:** Weight % of the different constituents of the foam formulations–3 different tannin-types for 3 different levels of surfactant concentration (1, 5, 10%).

Tannin-Type	Tannin [%]	FOH [%]	DEG [%]	H_2_O [%]	H_2_SO_4_ [%]	Tween80 [%]	S/T-F [%]
Mimosa, Quebracho, Chestnut	37.93	23.52	11.38	11.38	15.17	0.61	**1**
	37.02	22.95	11.11	11.11	14.81	3.00	**5**
	35.95	22.29	10.78	10.78	14.38	5.82	**10**

## Data Availability

The data of this study are available from the corresponding author.

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
