# Peer review of "Comparing Condensed and Hydrolysable Tannins for Mechanical Foaming of Furanic Foams: Synthesis and Characterization"

_molecules, 2023, doi:10.3390/molecules28062799_

Round 1

Reviewer 1 Report

The manuscript entitled “Mechanically Frothed Condensed and Hydrolysable Tannin Furanic Foams: Synthesis and Characterization” reported series of tannin-furanic foam by using two kinds of condensed tannin and a hydrolysable tannin, respectively. The authors provided some characterizations on them, such as the density, compression strength, FTIR, and NMR etc. In addition, the manuscript gave an integrated description to support the point of authors. But author need to modify and correct the manuscript carefully before the editor consider publishing it.

The detail comments are:

Introduction

-The abbreviation, XPS and EPS, should provide the full title when they first appear in the manuscript.

2.2.1 Foam synthesis

-This a universal problem in this manuscript throughout, some necessary latter space should be reserved, for example, Line 93, 114, 131, 263, etc.; for the units of ℃ and %, this kind od space should be removed. Please check the manuscript carefully.

-For the dimensions of the foam sample in the line 94, 15x15x5cm3 and 30x30x5cm3 are right?

-For the Table 1, this table did not express the experiments correctly. For example, the authors want to express that for each tannin material was used to prepared three kinds of foam samples, respectively. Or in the each formulation, the authors utilized all of them?

- Line 137, please revise the unit.

- Line 143, what is the “25C”?

Result and discussion

- Line 190, do the authors missed the unit of 30x30?

- The referee noticed that the authors reported the acid recovery and leaching. So, is the relationship between them? The acid recovery was calculated by the pH of leached water, but the leaching lower than the acid recovery in the sample of Quebracho and Chestnut.

- Please point out the peaks which were described in the manuscript at the Figure 6.

Author Response

Dear editor, dear, reviewer,

I would like to thank reviewer 1 for his/her time spent in reviewing our paper. A detailed answer to the raised comments can be found in the uploaded file.

Reviewer 2 Report

1 Abstract

The abstract should highlight the key results of the article.

2. Introduction

2.1 Line 34- “Tannin foams are highly porous materials, that can be obtained by acid/alkali catalysed polymerization of wood extractives with different crosslinkers”. Tannin is an extract of bark, not wood.

2.2 Several condensed tannins, such as quebracho tannin and mimosa tannin, and the foams based on these tannin and furfuryl alcohol have been studied and prepared, and many modifiers have been added later, such as nano cellulose to prepare high-strength foam. Several parameters are different in the various tannin foam formulations found in previous research to reduce costs and increase the mechanical and thermal properties. Therefore, good results have also been achieved. The innovation of this article is insufficient. The amount of furfuryl alcohol is still large and the price is still high. Moreover, hydrolyzed tannins are expensive, so why use them to prepare foams?

2.3 The author should make assumptions in the introduction, and then verify them in the results and discussions.

3. Materials and Method

3.1 In Table 1, it should be the Mimosa foam, Quebracho foam, Chestnut foam, and these three names should correspond to the data in the table.

3.2 In Table 1, why the amount of two condensed tannins is different, and the amount of other additives in TF foam is also different? Please explain!

3. Results and discussion

3.1 Line-153, “using 2 mol/L sulphuric acid in combination for condensed tannins works fine, while for chestnut the foam collapsed before curing due to the different reactivity of condensed and hydrolysable tannin”. There are many reasons for foam collapse, and it may also be the role of foaming agent. Please explain further here.

3.2 Line-188, “In this context, it should also be noted that the use of a larger container while retaining the same stirrer head led to higher densities in the production of the 30x30 foams with similar formulations “. Does this result tell us that using larger containers in the factory will result in different quality foam products?

3.3 Please indicate the names of samples corresponding to different positions in Figure 2.

3.4 Is there no standard deviation for each test value in Figure 3?

3.5 Line-233 “The highest compressive strength can be observed for mimosa while also having the highest elasticity”. Please explain the reason for this result.

3.6 No new discovery has been found in the result of of 13C-NMR and ATR-FT-IR. The reaction of Mimosa tannin and Quebracho tannin with furfuryl alcohol has been found in previous literature, while the surfactant and plasticizer added in the foam system have not been mentioned in this result.

4. Conclusions

The conclusion part is written too much, a little confused. It needs to be condensed into important and meaningful results

5. References

Reduce the self-citation rate to 15%.

Author Response

Dear editor, dear, reviewer,

I would like to thank reviewer 2 for his/her time spent in reviewing our paper. A detailed answer to the raised comments can be found in the uploaded file.

Round 2

Reviewer 2 Report

The author answers and modifies relevant questions, and the article can be accepted